# Acceptability of self-sampling human papillomavirus test for cervical cancer screening in Japan: A questionnaire survey in the ACCESS trial

Misuzu Fujita[1,2]*, Kengo Nagashima[3,4], Minobu Shimazu[5], Misae Suzuki[6], Ichiro Tauchi[6], Miwa Sakuma[6], Setsuko Yamamoto[6], Hideki Hanaoka[5], Makio Shozu[7], Nobuhide Tsuruoka[8], Tokuzo Kasai[1], Akira Hata[1,9]

1 Department of Health Research, Chiba Foundation for Health Promotion and Disease Prevention, Chiba, Japan, 2 Department of Public Health, Chiba University Graduate School of Medicine, Chiba, Japan, 3 Biostatistics Unit, Clinical and Translational Research Center, Keio University Hospital, Shinjuku-ku, Tokyo, Japan, 4 Research Center for Medical and Health Data Science, The Institute of Statistical Mathematics, Tachikawa, Tokyo, Japan, 5 Clinical Research Center, Chiba University Hospital, Chiba, Japan, 6 Municipal Health Center, Department of Health and Welfare, Ichihara City, Ichihara, Chiba, Japan, 7 Departments of Reproductive Medicine, Chiba University Graduate School of Medicine, Chiba, Japan, 8 Yushudai Clinic, Ichihara, Chiba, Japan, 9 Center for Preventive Medical Sciences, Chiba University, Chiba, Japan

* mi-hujita@kenko-chiba.or.jp

**Data Availability Statement:** All relevant data are within the Supporting Information File (S1 Data).

## Abstract

### Purpose

In terms of medical policy for cervical cancer prevention, Japan lags far behind other industrialized countries. We initiated a randomized controlled trial to evaluate the self-sampling human papillomavirus (HPV) test as a tool to raise screening uptake and detection of pre-cancer. This study was conducted to explore the acceptability and preference of self-sampling using a subset of the data from this trial.

### Methods

A pre-invitation letter was sent to eligible women, aged 30−59 years who had not undergone cervical cancer screening for three or more years. After excluding those who declined to participate in this trial, the remaining women were assigned to the self-sampling and control groups. A second invitation letter was sent to the former group, and those wanting to undergo the self-sampling test ordered the kit. A self-sampling HPV kit, consent form, and a self-administered questionnaire were sent to participants who ordered the test.

### Results

Of the 7,340 participants in the self-sampling group, 1,196 (16.3%) administered the test, and 1,192 (99.7%) answered the questionnaire. Acceptability of the test was favorable; 75.3−81.3% of participants agreed with positive impressions (easy, convenient, and clarity of instruction), and 65.1−77.8% disagreed with negative impressions (painful, uncomfortable, and embarrassing). However, only 21.2% were confident in their sampling procedure.

**Funding:** This work was supported by Japan Society for the Promotion of Science (JSPS) KAKENHI Grant (AH, Number 20H03906, http://www.jsps.go.jp/j-grantsinaid). The funder had no role in the study design, writing the report, or the decision to submit the report for publication.

**Competing interests:** The authors have declared that no competing interests exist.

Willingness to undergo screening with a self-collected sample was significantly higher than that with a doctor-collected sample (89.3% vs. 49.1%; p<0.001). Willingness to undergo screening with a doctor-collected sample was inversely associated with age and duration without screening (both p<0.001), but that with a self-collected sample was not associated.

## Conclusions

Among women who used the self-sampling HPV test, high acceptability was confirmed, while concerns about self-sampling procedures remained. Screening with a self-collected sample was preferred over a doctor-collected sample and the former might alleviate disparities in screening rates.

## Introduction

Cervical cancer is caused by a persistent high-risk human papillomavirus (HPV) infection and can be prevented by the adequate implementation of both HPV vaccination and organized screening programs [1, 2]. However, in Japan, the Ministry of Health, Labour and Welfare suspended active HPV vaccine recommendations in 2013 and the vaccination rate dropped to almost 0% in 2015 [3, 4]. The recommendations were reinstated in April 2022; however, the negative effects of the suspended recommendations remain. Additionally, compared to other developed countries, the screening rate in Japan is extremely low [5]. Likely due to these factors, cervical cancer morbidity and mortality have recently increased [6]. As the strategy to prevent cervical cancer depends greatly on the screening program, improving the screening rate is an urgent concern in Japan.

Cervical cytology has been widely implemented worldwide as a cervical cancer screening modality, which cannot be performed on a self-collected sample because of possible failure in securing a sufficient number of cervical cells. Therefore, this conventional screening needs to be performed at a medical institution to enable the collection of the sample by a doctor, and thus, involves physical and psychological barriers, such as a lack of time, embarrassment, discomfort, and fear [7–11]. In 2020, the guidelines for cervical cancer screening published by the National Cancer Center Japan were revised, and they recommended HPV testing as a primary screening method for the first time. Unlike cytology, HPV testing can be performed on a self-sampled specimen because of the strong concordance of HPV detection between physician-collected and self-collected samples [12–14]; additionally, there is similar accuracy in the detection of cervical intraepithelial neoplasia (CIN) grade 2 or worse and grade 3 or worse [15–17]. HPV testing using a self-sampled specimen can overcome the barriers in conventional screening using cytology. Meta-analyses indicated that self-sampling HPV testing increased screening uptake [17, 18] and resulted in increased detection of CIN 2 or worse [17]. Therefore, the self-sampling HPV test has been implemented as an alternative option for conventional screening non-responders in several countries [19, 20]. However, the guidelines in Japan do not recommend the use of self-collected samples for HPV testing because of insufficient evidence of the effectiveness and feasibility in the country. Therefore, robust evidence of the test's effectiveness is strongly sought.

In line with this, we initiated a randomized controlled trial in 2020, the Accelerating Cervical Cancer Elimination by Self-Sampling test (ACCESS) trial, to evaluate the effectiveness of the self-sampling HPV test in cervical cancer screening uptake and pre-cancer detection. We conducted a pre-planned questionnaire survey as a part of this trial to examine the main

reasons for not undergoing cervical cancer screening, knowledge about HPV, acceptability of the self-sampling HPV test, and screening preference among Japanese women. The opinions based on the experience of self-sampling in the target population for cervical cancer screening will provide meaningful information to implement the test as a practice.

## Materials and methods

### Participants

This study is a questionnaire survey linked with the ACCESS trial, an ongoing randomized controlled trial. The trial is registered at the Japan Registry of Clinical Trials (jRCT, 1030200276), and the study protocol including the statistical analysis plan has been published [21]. The sample size was determined based on the primary endpoint in the ACCESS trial. A flowchart of this study is presented in Fig 1.

A total of 20,555 women who met the inclusion criteria were extracted from the database of Ichihara City Hall on December 22, 2020, and a pre-invitation letter was sent to them on February 1, 2021. Inclusion criteria were 1) women living in Ichihara City as of December 22, 2020; 2) women aged 30–59 years as of April 1, 2021; 3) the target population for cervical cancer screening provided by Ichihara City in 2021—that is, women whose age was an even number (the city set this eligibility requirement to offer cervical cancer screening once every two years to all women); and 4) women who had not received routine cervical cancer screening provided by Ichihara City for three or more years. The fourth inclusion criterion was established because we assumed that if implemented in Japan, the self-sampling HPV test would be used by routine screening non-responders as in other countries where the test has been implemented [19, 20]. The pre-invitation letter indicated that the women can refuse to participate in the trial (opt out) and the procedures. Of the 20,555 women who met the inclusion criteria, 12 women whose pre-invitation letter was returned owing to an incorrect address and 4,283 women who opted out prior to February 22, 2021, were excluded, and the remaining 16,260 women were assigned randomly to the self-sampling (N = 8,145) and control groups (N = 8,115) at a 1:1 ratio according to computer-generated numbers. Participants assigned to the self-sampling group could receive a cytology test or screening with a self-sampling HPV test of their own free will, while those in the control group could receive a cytology test. For participants assigned to the self-sampling group, a second invitation letter was sent on March 10, 2021. The letter indicated that they could use a self-sampling HPV test and provided instructions to order the test. For participants who ordered the test by June 30, 2021, a self-sampling kit (Evalyn® Brush, Rovers® Medical Devices, Netherland), an instruction manual explaining how to take a sample, a booklet explaining how to send a sample, an informed consent form, a questionnaire, and a return addressed envelope with cash on delivery were sent. The participants of this questionnaire survey were those in the self-sampling group who submitted three items by September 3, 2021: the completed consent form, self-collected sample, and completed questionnaire. Some of the authors had access to information that could identify individual participants during and after data collection to administrate the data for this trial.

### Questionnaire

The questionnaire was developed with reference to previous studies [7, 8, 22–27] and is available elsewhere [21]. As reasons for not undergoing cervical cancer screening provided by Ichihara City, 15 items were presented to the participants, and they could select all applicable options. Experience of self-sampling was examined with eight items: five for acceptability that were related to pain, discomfort, embarrassment, ease, and convenience; two for clarity of instruction; and one for confidence in self-sampling. Responses were given on a five-point

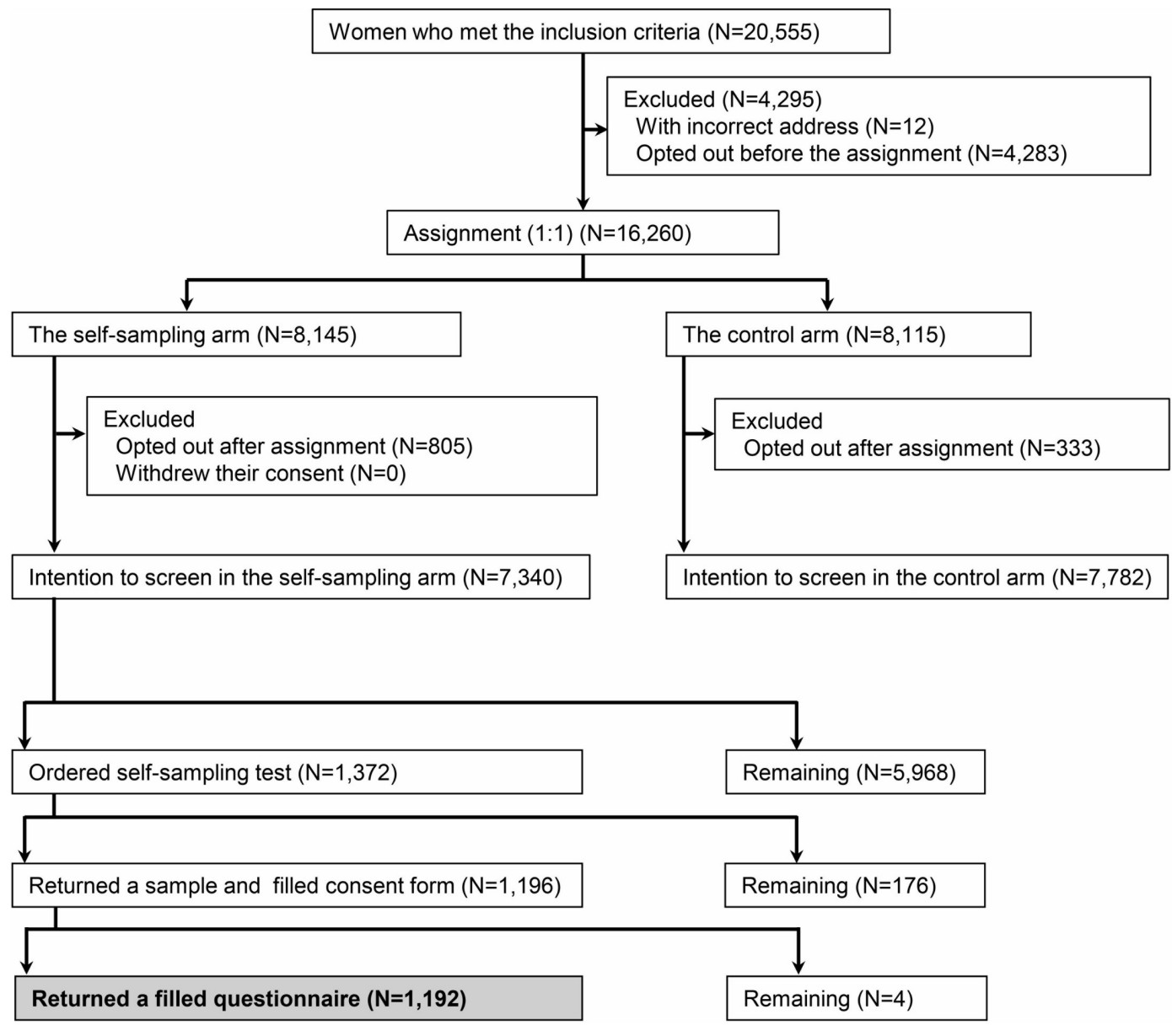

**Fig 1. Study flowchart.** Participants of this study are indicated in the gray cell.

Likert scale: "Fully disagree," "Somewhat disagree," "Neither agree nor disagree," "Somewhat agree," and "Fully agree." To examine the women's knowledge about HPV before and after participating in this trial, the following questions were asked: "Did you know that HPV is the cause of cervical cancer before participating in this trial?" and "Do you know that HPV is the cause of cervical cancer now?" Participants could select one of two options: "I did not know (I don't know)" or "I knew (I know)." To examine cervical cancer screening preference, we asked the following questions: "Will you receive cervical cancer screening in the future via sampling conducted by a doctor?" and "Will you receive cervical cancer screening in the future via sampling conducted by yourself?" Participants could select one of the following three options for each condition: "I will receive," "I will not receive," or "Unknown."

## Other data

Participants' age as of April 1, 2021, and the latest year to have undergone cervical cancer screening provided by Ichihara City were extracted from the database of the Ichihara City Hall and provided for this trial. Using the latter, the duration without screening was calculated. Age was categorized as 30–39, 40–49, and 50–59 years. The duration without screening was categorized as 3–5 years, 6 years or more, and no screening records.

## Statistical analysis

Statistical analyses were performed using STATA software version 16.0 (STATA LP, College Station, TX). The database was locked to this questionnaire survey on October 6, 2021. A comparison between the questionnaire participants and questionnaire non-responders was performed using the student's t-test and chi-squared test. Answers to the questions related to screening preference were converted to a binary: "I will receive" and "others" (including "I will not receive" and "Unknown"). For binary data, frequency, percentage, and 95% confidence intervals were calculated. To compare knowledge about HPV before and after participating in this trial, McNemer's test was conducted. McNemer's test was also performed to compare the willingness to undergo two hypothetical screenings—screening with a doctor-collected sample and that with a self-collected sample. As a sensitivity analysis, the willingness was compared excluding participants who selected "I had an opportunity to undergo cytology testing other than the screening provided by the city" as the reason for not undergoing screening. As post-determined analyses, experience of self-sampling, knowledge about HPV, and willingness to undergo screening were compared with age and duration without screening using the chi-squared test and logistic regression analysis. After calculating the estimates using logistic regression analysis, a linear trend test was performed using the "contrast p." command of STATA. For this analysis, items of self-sampling experience were converted into a binary: "Agree" (including "Fully agree" and "Somewhat agree") and "Others" (including "Neither agree nor disagree," "Somewhat disagree," and "Fully disagree"). Missing values were included in the aggregate but excluded from the statistical test. A two-tailed p-value of $<0.05$ was considered significant.

## Ethics statement

The questionnaire survey is a secondary analysis of the ACCESS trial. Written consent was obtained from the participants who underwent HPV testing and completed the questionnaire survey. The details have already been published [21]. The trial was approved by the Research Ethics Committees of the Chiba Foundation for Health Promotion and Disease Prevention (approval numbers R2-2 and R2-7), Graduate School of Medicine, Chiba University (approval number 3979), and the Institute of Statistical Mathematics (approval number ISM20-001) and conducted in accordance with the Declaration of Helsinki and the Ethical Guidelines for Medical and Health Research Involving Human Subjects.

## Results

Of the 7,340 participants in the self-sampling group, 1,372 (18.7%) ordered the self-sampling HPV test, and 1,196 (16.3%) returned both the completed consent form and a self-collected sample. Of those, 1,192 (99.7%) also returned the completed questionnaire as shown in Fig 1. Characteristics of the questionnaire survey participants and questionnaire non-responders are shown in Table 1. Age and duration without screening were significantly different between the two.

The most common reasons for not undergoing screening were a lack of time to undergo screening (45.4%), followed by being too bothered to make an appointment (44.7%), embarrassment (24.7%), having an opportunity to undergo cytology other than the screening provided by the city (23.6%), and pain or discomfort (17.8%) as shown in Fig 2. Zero participants selected hysterectomy as the reason, likely because we instructed participants who had received a hysterectomy to not order the self-sampling HPV test in the second invitation letter.

Impressions of the self-sampling HPV test based on experience are detailed in Fig 3. More than 60% of the participants answered "Fully disagree" or "Somewhat disagree" for the items related to negative impressions of the sampling, such as painful, uncomfortable, and embarrassing. More than 70% answered "Fully agree" or "Somewhat agree" for the items related to positive impressions, such as easy and convenient. Additionally, most participants felt that the user instructions were clear and the device worked as the instructions explained. However, most participants did not have confidence in their sample collection; only 21.2% answered "Fully agree" or "Somewhat agree" to the item "I believe I was successful in collecting the sample."

The association of sampling experience with age and duration without screening are shown in S1–S8 Tables in S1 File. Older women more frequently felt that the sampling was painful (p for trend = 0.002) and less frequently that the sampling was easy (p for trend = 0.030) compared with younger women.

Knowledge about HPV significantly increased after participating in this trial (46.9% vs. 86.7%; p<0.001) as shown in Table 2. There were no associations between age and knowledge before and after participation. In contrast, longer duration without screening was significantly associated with less knowledge about HPV before participation (p for trend = 0.027), but not after (p for trend = 0.291).

Willingness to undergo screening with self-collected samples was significantly higher than that with doctor-collected samples (89.3% vs. 49.1%; p<0.001) as shown in Table 3. Additionally, older age and longer duration without screening were associated with lower willingness to undergo screening with doctor-collected samples (p for trend <0.001 for both), but there were no such associations with self-collected samples (p for trend = 0.182 and 0.702, respectively). After excluding those who answered "I had an opportunity to undergo cytology testing other than the screening provided by the city" as the reason for not undergoing screening, similar results were observed, as shown in S9 Table in S1 File.

## Discussion

In this study, which was a secondary analysis of the ACCESS trial, we revealed the main reasons for not undergoing cervical cancer screening, the degree of knowledge about HPV before and after participating in this trial, the acceptability of self-sampling, and future screening preference.

Previous studies reported that the main reasons for not having a cytology test were practical barriers, such as a lack of time to undergo the test and forgetting to schedule an appointment, and emotional barriers, such as embarrassment, discomfort, and pain [7–11]. This study confirmed similar reasons in a relatively large sample of Japanese women for the first time. The reasons for not undergoing screening are common across countries and most barriers might be resolved through implementation of the self-sampling test. Meanwhile, a reason specific to Japan was also observed: "having an opportunity to undergo cytology testing other than the screening provided by the city," which was the fourth most frequent reason. In Japan, cervical cancer screening is performed as organized screening, which is generally provided by municipalities and companies, or opportunistic screening. Unlike other countries, such as the Netherlands, Japan does not have a registry system for cervical cancer screening. Therefore, we could

**Table 1. Characteristics of the questionnaire survey participants and non-responders.**

|  | Participants | Non-responders | p-value |
|---|---|---|---|
| Number | 1,192 | 6,148 |  |
| Age (years) |  |  |  |
| Mean (Standard deviation) | 44.1 (8.2) | 44.7 (8.4) | 0.013 [b] |
| Age category [a] |  |  |  |
| 30−39 years | 384 (32.2) | 1,788 (29.1) | 0.001 [c] |
| 40−49 years | 451 (37.8) | 2,186 (35.6) |  |
| 50−59 years | 357 (30.0) | 2,174 (35.4) |  |
| Duration without screening [a] |  |  |  |
| 3−5 years | 203 (17.0) | 570 (9.3) | <0.001 [c] |
| 6 years or more | 279 (23.4) | 1,188 (19.3) |  |
| No screening records | 710 (59.6) | 4,390 (71.4) |  |

Participants were those who returned the completed consent form, self-collected sample, and completed questionnaire.

[a] Number (%)

[b] Student's t-test

[c] Chi-square test

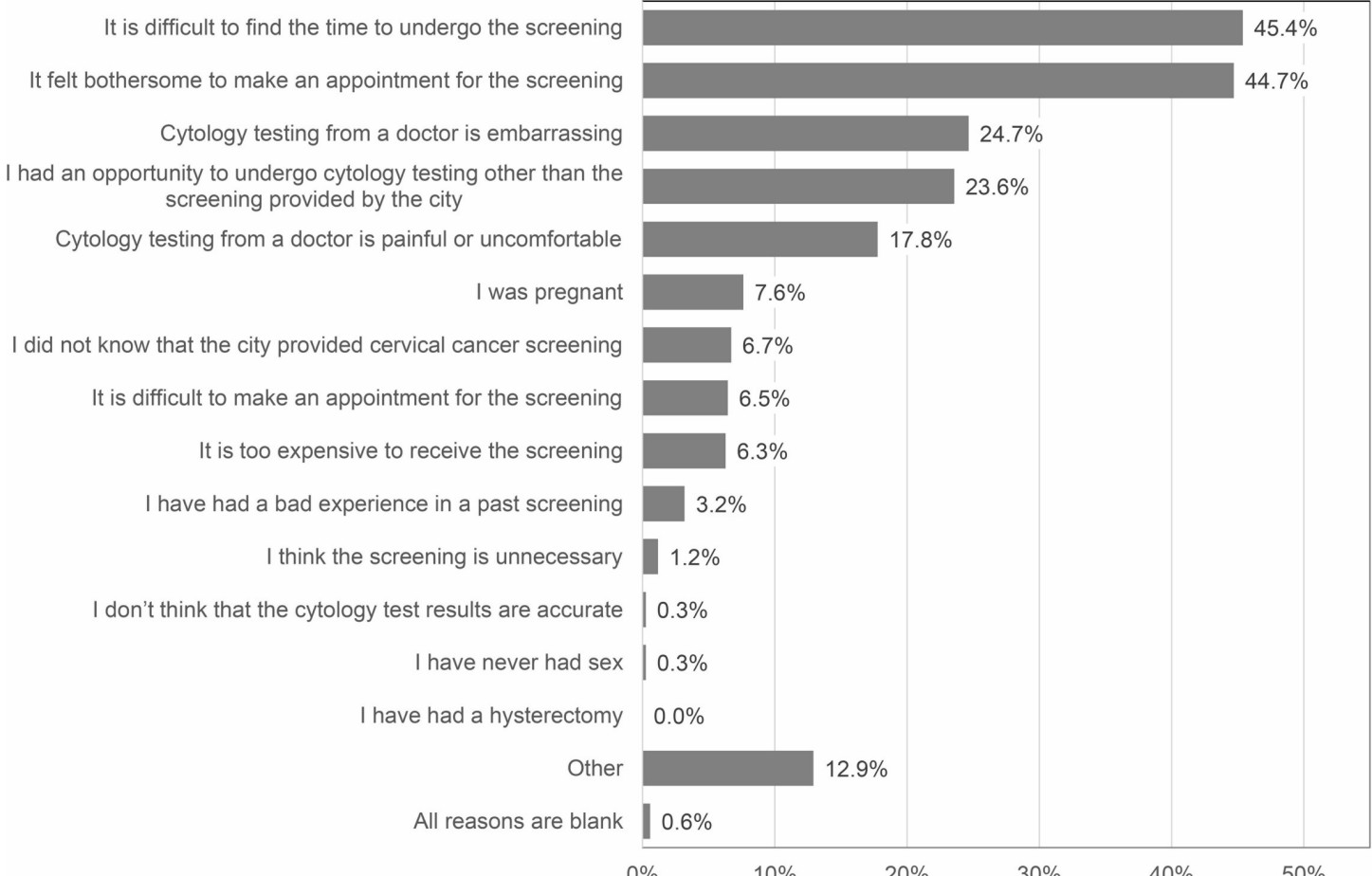

**Fig 2. Reasons for not undergoing cervical cancer screening provided by the city.** The number of subjects was 1,192. Participants could select all applicable reasons.

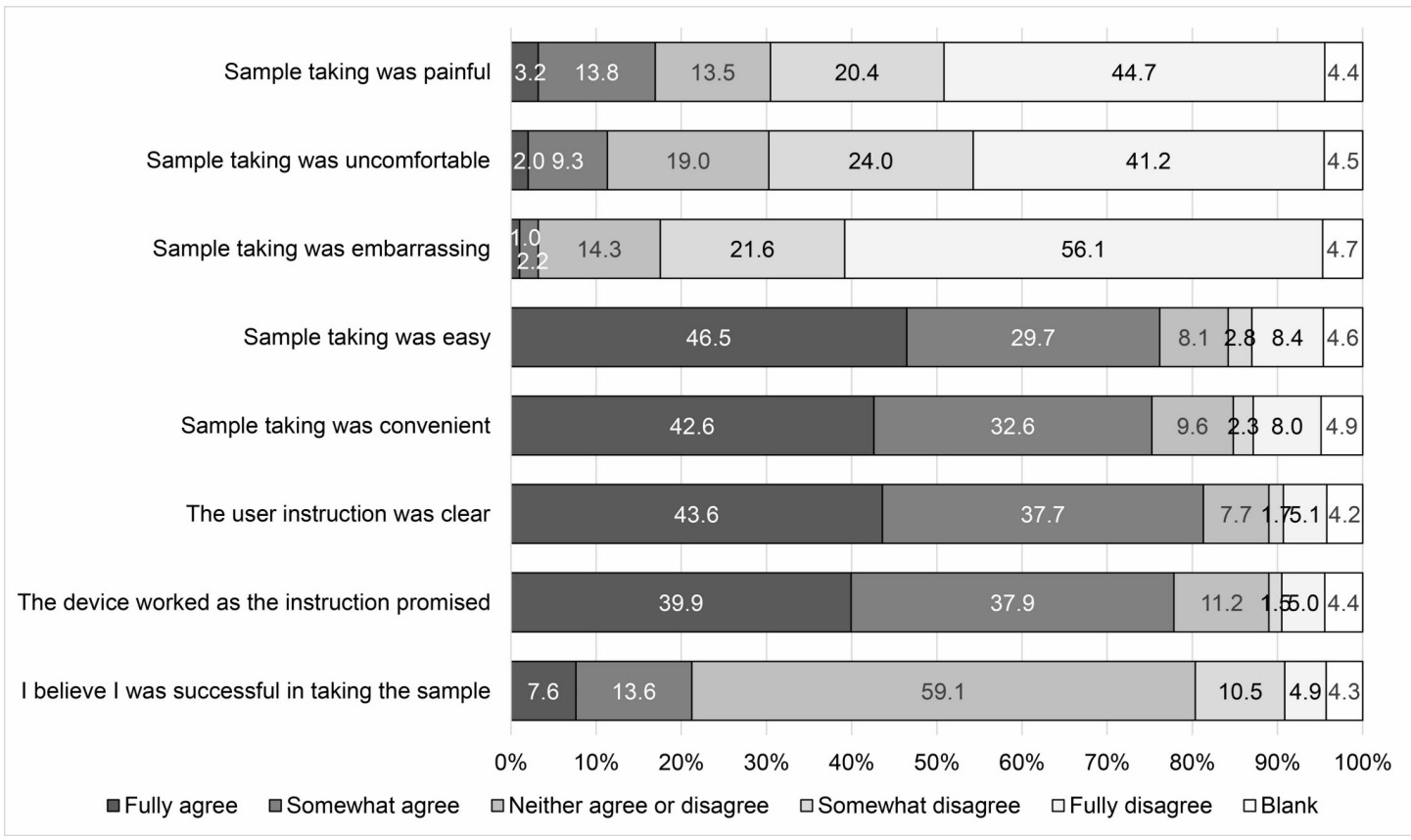

**Fig 3. Impressions of the self-sampling HPV test based on experience.**

not truly determine screening non-responders and under-responders. Although we extracted women who did not undergo cervical cancer screening for three or more years from Ichihara City Hall's database, which is the most reliable method of extraction, some participants had an opportunity to undergo a cytology test provided by other sites.

Acceptability of the self-sampling HPV test was favorable. This result is consistent with many previous studies across various countries [7, 22–28]. In contrast to the overall favorable acceptability of the test, concerns about the self-sampling procedures were recognized. Although a simple comparison cannot be made because of methodological differences, the proportion of participants who had confidence in their sampling seemed to be lower than that reported by previous studies in Australia [7], the Netherlands [22, 24], Hong Kong [29], and Finland [25, 26]. One reason might be that the test was not familiar to Japanese women, because the self-sampling HPV test has not been recommended by official guidelines and has been rarely implemented in Japan. We recently reported that the self-sampling HPV test rarely yielded invalid results among Japanese women [30], which is consistent with previous studies [10, 31–35]. Additionally, the results of HPV testing using a self-collected sample have strong concordance with a doctor-collected sample [12–14], and similar accuracy in the detection of CIN grade 2 or worse and grade 3 or worse between self-collected and doctor-collected samples was also confirmed [15–17]. To eliminate women's concerns related to the accuracy of the self-sampling HPV test, informing them of these facts in advance would be useful, especially when they undergo the test for the first time.

**Table 2. Knowledge about HPV before and after participation in this trial.**

| | All | N [d] | Percent (95% CI) | OR (95% CI) | p-value |
|---|---|---|---|---|---|
| Before participation in this trial | | | | | |
| All | 1,156[a] | 542 | 46.9 (44.0−49.8) | − | <0.001[e] |
| Age | | | | | |
| 30−39 years | 381[b] | 171 | 44.9 (39.8−50.0) | 1.00 | 0.046[f] |
| 40−49 years | 446[b] | 231 | 51.8 (47.0−56.5) | 1.32 (1.00−1.74) | 0.784[g] |
| 50−59 years | 351[b] | 154 | 43.9 (38.6−49.2) | 0.96 (0.72−1.29) | |
| Duration without screening | | | | | |
| 3−5 years | 201[b] | 108 | 53.7 (46.6−60.8) | 1.00 | 0.077[f] |
| 6 years or more | 275[b] | 133 | 48.4 (42.3−54.4) | 0.81 (0.56−1.16) | 0.027[g] |
| No screening records | 702[b] | 315 | 44.9 (41.1−48.6) | 0.70 (0.51−0.96) | |
| After participation in this trial | | | | | |
| All | 1,156[a] | 1,002 | 86.7 (84.6−88.6) | − | − |
| Age | | | | | |
| 30−39 years | 376[c] | 316 | 84.0 (79.9−87.6) | 1.00 | 0.112[f] |
| 40−49 years | 438[c] | 390 | 89.0 (85.7−91.8) | 1.54 (1.03−2.32) | 0.345[g] |
| 50−59 years | 342[c] | 296 | 86.5 (82.5−90.0) | 1.22 (0.81−1.85) | |
| Duration without screening | | | | | |
| 3−5 years | 197[c] | 174 | 88.3 (83.0−92.5) | 1.00 | 0.273[f] |
| 6 years or more | 269[c] | 239 | 88.8 (84.5−92.3) | 1.05 (0.59−1.88) | 0.291[g] |
| No screening records | 690[c] | 589 | 85.4 (82.5−87.9) | 0.77 (0.48−1.25) | |

CI: confidence interval; OR: odds ratio

[a] Number of participants who answered both questions related to knowledge about HPV before and after participation in this trial. Missing values were 36.

[b] Number of participants who answered a question related to knowledge about HPV before participation in this trial. Missing values were 14.

[c] Number of participants who answered a question related to knowledge about HPV after participation in this trial. Missing values were 36.

[d] Number of participants who answered "I knew" or "I know."

[e] Comparison of knowledge before and after participation in this trial in all participants using McNemer's test.

[f] Association of knowledge with age or duration without screening using chi-squared test.

[g] Association of knowledge with age or duration without screening using linear trend test.

Knowledge about HPV increased after participating in this trial, suggesting that experience of the self-sampling HPV test improved knowledge. We also found a significant association between longer duration without screening and less knowledge before participating in this trial; however, the association disappeared after participation. Previous studies reported evidence of disparity with regard to women's knowledge about HPV; the knowledge was less among women whose cervical cancer mortality was disproportionately high, including racial and ethnic minorities and those with low socioeconomic status [36, 37], who had lower cervical cancer screening rate in general [38, 39]. This study's results suggest that experience of the test can not only improve knowledge but also reduce disparities related to knowledge.

As for future screening preference, screening with self-collected samples was preferred over that with doctor-collected samples, which is consistent with previous studies [7, 22, 23]. Unexpectedly, 49.1% of participants were willing to undergo future screening even if samples would be collected by a doctor. Given that the participants were limited to those who had not undergone screening for three years or more, this value is higher than expected. However, this might be overestimated, because we could not extract true cervical screening non-responders and under-responders owing to a lack of a registry system of cervical cancer screening. Thus, as a sensitivity analysis, participants who selected "I had an opportunity to undergo cytology

**Table 3. Willingness to undergo screening per type of sample collection.**

| | All | N [d] | Percent (95% CI) | OR (95% CI) | p-value |
|---|---|---|---|---|---|
| Screening with doctor-collected samples | | | | | |
| All | 1,126 [a] | 553 | 49.1 (46.2–52.1) | – | <0.001 [e] |
| Age | | | | | |
| 30−39 years | 371 [b] | 216 | 58.2 (53.0–63.3) | 1.00 | <0.001 [f] |
| 40−49 years | 429 [b] | 220 | 51.3 (46.4–56.1) | 0.76 (0.57−1.00) | <0.001 [g] |
| 50−59 years | 337 [b] | 124 | 36.8 (31.6–42.2) | 0.42 (0.31−0.57) | |
| Duration without screening | | | | | |
| 3−5 years | 199 [b] | 123 | 61.8 (54.7–68.6) | 1.00 | <0.001 [f] |
| 6 years or more | 261 [b] | 136 | 52.1 (45.9–58.3) | 0.67 (0.46−0.98) | <0.001 [g] |
| No screening records | 677 [b] | 301 | 44.5 (40.7–48.3) | 0.49 (0.36−0.68) | |
| Screening with self-collected samples | | | | | |
| All | 1,126 [a] | 1,005 | 89.3 (87.3–91.0) | – | |
| Age | | | | | |
| 30−39 years | 377 [c] | 343 | 91.0 (87.6–93.7) | 1.00 | 0.356 [f] |
| 40−49 years | 441 [c] | 390 | 88.4 (85.1–91.3) | 0.76 (0.48−1.20) | 0.182 [g] |
| 50−59 years | 348 [c] | 306 | 87.9 (84.0–91.2) | 0.72 (0.45−1.16) | |
| Duration without screening | | | | | |
| 3−5 years | 201 [c] | 180 | 89.6 (84.5–93.4) | 1.00 | 0.771 [f] |
| 6 years or more | 273 [c] | 246 | 90.1 (85.9–93.4) | 1.06 (0.58−1.94) | 0.702 [g] |
| No screening records | 692 [c] | 613 | 88.6 (86.0–90.9) | 0.91 (0.54−1.51) | |

CI: confidence interval; OR: odds ratio

[a] Number of participants who answered both questions related to willingness to undergo screening with doctor-collected samples and self-collected samples. Missing values were 66.

[b] Number of participants who answered a question related to willingness to undergo screening with doctor-collected samples. Missing values were 55.

[c] Number of participants who answered a question related to willingness to undergo screening with self-collected samples. Missing values were 26.

[d] Number of participants who answered, "I will receive."

[e] Comparison of willingness between two screenings using McNemer's test.

[f] Comparison of willingness with age or duration without screening using chi-squared test.

[g] Comparison of willingness with age or duration without screening using linear trend test.

testing other than the screening provided by the city" as the reason for not undergoing the screening were excluded. As a result, 43.5% of women remained willing. Experience with the self-sampling HPV test may strengthen the willingness to undergo a cytology test, which requires a doctor-collected sample, over practical and emotional barriers.

Additionally, older age and longer duration without screening were associated with a lower willingness to undergo screening with doctor-collected samples, but these associations were not observed in the screening with self-collected samples. In all subgroups, approximately 90% of participants were willing to undergo screening with self-collected samples. A comparison between age and impressions of the self-sampling test revealed that older women more frequently experienced pain and were less likely to feel that self-sampling was easy compared with younger women. Nonetheless, most older women seemed to prefer the self-sampling method. These results suggest that the self-sampling HPV test is an acceptable method regardless of age and the duration without screening and may result in reducing disparities in screening rates.

A major limitation of this study is the low proportion of participants who underwent the self-sampling HPV test, that is 16.3% of the eligible women. This affects the interpretation of

the results. First, even if the acceptability of the test is favorable, under the assumed situation that the screening uptake in the self-sampling group is lower than that in the control group, the self-sampling HPV test is useless. This trial is ongoing. Therefore, conclusions about the effectiveness of the self-sampling HPV test must await the results of the primary endpoint. However, given that the subjects of this trial were women who had not received cervical cancer screening for three or more years, this participation rate is thought to be meaningful. Additionally, the previous studies observed similar participation rates [40–45] and reported that the self-sampling HPV test contributed to improving the screening uptake [40–42, 44, 45]. Second, selection bias might be induced. As the subjects of this questionnaire survey were women who underwent the self-sampling HPV test at their own will and the participation rate was 16.3%, the subjects may be biased toward those who had a positive impression of the test. As a result, favorable acceptability and a strong preference for self-sampling might be overestimated compared to the general population. Indeed, significant differences were observed in the characteristics of the questionnaire participants and questionnaire non-responders. However, relatively low participation rates for a self-sampling test were observed in previous studies, which revealed the acceptability of the test [8, 25, 27, 28]. This is inevitable to establish the subject's autonomous participation. The current study revealed high acceptance and preference for the self-sampling HPV test among women who experienced the test, not in the general population. It is a strength of the study that almost all the participants who underwent self-sampling answered the questionnaire (99.7%), which was similar to or higher than the previous studies [8, 23, 25, 27, 28]. In Japan, women are not familiar with a self-sampling HPV test. Therefore, most participants experienced the test for the first time. The following descriptions were found frequently in the questionnaire's free text box; "I thought it was difficult to take a sample by myself at first, when I tried, it was easy." This suggests that the initial impressions changed with experience. Although there would be women with negative impressions of the self-sampling HPV test in the general population, such women might also change their minds through experience. Providing opportunities for women to perform the self-sampling test, even on a trial basis, would be useful in Japan.

In conclusion, this study reveals high acceptability and preference for the self-sampling HPV test among women who used the test and adds further evidence to strengthen a hypothesis that the test is acceptable to women across cultures and countries. Additionally, this study suggests that the self-sampling HPV test reduces disparities in knowledge about HPV and screening rates. Thus, this study provides useful evidence to decision-makers to implement the self-sampling HPV test as a practice in Japan, where cervical cancer prevention measures are seriously lacking.

## Supporting information

**S1 File.**
(DOCX)

**S1 Data.**
(XLS)

**S1 Checklist. STROBE statement—checklist of items that should be included in reports of observational studies.**
(DOCX)

## Acknowledgments

We sincerely appreciate the invaluable help and support of the mayor of Ichihara City Hall, Joji Koide. We also sincerely appreciate the support of the president of the Chiba Foundation for Health Promotion and Disease Prevention, Takehiko Fujisawa, and the staff members, Chiori Suzuki, Fumika Kumahara, Fumiya Chiwaki, Hideaki Nagai, Ikumu Matsushita, Kenji Ishii, Makoto Koumi, Michiko Fusaeda, and Saeri Omori. We would like to thank Editage (www.editage.com) for English language editing.

## Author Contributions

**Conceptualization:** Misuzu Fujita, Kengo Nagashima, Minobu Shimazu, Misae Suzuki, Ichiro Tauchi, Miwa Sakuma, Setsuko Yamamoto, Hideki Hanaoka, Makio Shozu, Nobuhide Tsuruoka, Tokuzo Kasai, Akira Hata.

**Data curation:** Misuzu Fujita.

**Formal analysis:** Misuzu Fujita.

**Funding acquisition:** Akira Hata.

**Investigation:** Misuzu Fujita, Misae Suzuki, Ichiro Tauchi, Miwa Sakuma, Setsuko Yamamoto, Akira Hata.

**Methodology:** Misuzu Fujita, Kengo Nagashima, Minobu Shimazu, Misae Suzuki, Ichiro Tauchi, Miwa Sakuma, Setsuko Yamamoto, Hideki Hanaoka, Tokuzo Kasai, Akira Hata.

**Project administration:** Akira Hata.

**Supervision:** Minobu Shimazu, Hideki Hanaoka, Makio Shozu, Nobuhide Tsuruoka, Tokuzo Kasai.

**Validation:** Misuzu Fujita.

**Visualization:** Misuzu Fujita.

**Writing – original draft:** Misuzu Fujita.

**Writing – review & editing:** Misuzu Fujita, Kengo Nagashima, Minobu Shimazu, Misae Suzuki, Ichiro Tauchi, Miwa Sakuma, Setsuko Yamamoto, Hideki Hanaoka, Makio Shozu, Nobuhide Tsuruoka, Tokuzo Kasai, Akira Hata.

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
