## [Decision Letter · Decision Letter 0]

28 Dec 2022

PONE-D-22-28147Acceptability of self-sampling human papillomavirus test for cervical cancer screening in Japan: Questionnaire survey in the ACCESS trialPLOS ONE

Dear Dr. Fujita,

Thank you for submitting your manuscript to PLOS ONE. After careful consideration, we feel that it has merit but does not fully meet PLOS ONE’s publication criteria as it currently stands. Therefore, we invite you to submit a revised version of the manuscript that addresses the points raised during the review process.

You might want to focus on below listed revisions in addition to the reviewers’ comments:

1. English writing needs to be improved. Awkward or ambiguous English expressions and incorrect grammar are seen in current version, e.g. the 3^rd^ sentence in Methods of Abstract, the last sentence in Results of Abstract, ... The authors are advised please to check thoroughly the entire manuscript for language improvement, and/or ask help for English editing if available at your own end.

2. The current manuscript is unnecessarily lengthy. Please check the reviewers’ comments as well. Please write concisely in the revision with removing irrelevant sentences/paragraphs and duplicated/redundant materials.

3. Although the parenting study is an ongoing control randomized trial, however the current manuscript analyzed a subset of the data from one arm only. Please make this crystal clear, specially in abstract and here there when it needs to be clarified.

 4. Please look at Reviewer 2’s comment about the sampling bias concern. Please define your study population explicitly and give the limitation where your findings can be generalized to.

We look forward to receiving your revised manuscript.

Kind regards,

Ruofei Du, PhD

Academic Editor

PLOS ONE

Journal Requirements:

2. Our staff editors have determined that your manuscript is likely within the scope of our Early Detection, Screening and Diagnosis of Cancer Call for Papers. This editorial initiative is headed by in-house PLOS editors. This Call for Papers aims to explore recent advances in the early detection of cancer and implications of these advances for patient survival. Additional information can be found on our announcement page: https://collections.plos.org/call-for-papers/early-detection-screening-and-diagnosis-of-cancer/

If you would like your manuscript to be considered for this collection, please let us know in your cover letter and we will ensure that your paper is treated as if you were responding to this call. Please note that being considered for the Call for Papers does not require additional peer review beyond the journal’s standard process and will not delay the publication of your manuscript if it is accepted by PLOS ONE. If you would prefer to remove your manuscript from collection consideration, please specify this in the cover letter.

Reviewers' comments:

Reviewer's Responses to Questions

**Comments to the Author**

1. Is the manuscript technically sound, and do the data support the conclusions?

Reviewer #1: Yes

Reviewer #2: No

2. Has the statistical analysis been performed appropriately and rigorously? 

Reviewer #1: Yes

Reviewer #2: No

3. Have the authors made all data underlying the findings in their manuscript fully available?

Reviewer #1: Yes

Reviewer #2: Yes

4. Is the manuscript presented in an intelligible fashion and written in standard English?

Reviewer #1: Yes

Reviewer #2: Yes

5. Review Comments to the Author

Reviewer #1: The authors conducted a pre-planned questionnaire survey as a part of this trial to examine the main reasons for not undergoing cervical cancer screening, knowledge about HPV, acceptability of the self-sampling HPV test, and screening preference in the future in Japanese women.

They show that the acceptability of the self-sampling HPV test was confirmed, while concerns about self-sampling procedures remained.

They also suggested Screening with a self-collected sample might alleviate disparities in screening rates. This study is well conducted, and the methods used are appropriate. The data is presented clearly. These findings will be of interest to obstetrical practitioners, as well as researchers in the field.

I would like to express my homage to the authors for completing a valuable study and for the extensive analysis. I am convinced that this manuscript has an important message to obstetrical clinicians. This manuscript can be considered positively for publication, but I suggest the authors revise the following points to improve this manuscript for publication.

1. I have a question about patient selection criteria. Please explain briefly why you excluded patients who received regular cervical cancer screening and included only those who had not been screened for three years or more. We believe that asking patients who are regularly screened for their opinion on self-sampling is also helpful and has the advantage of reducing selection bias.

2. It is important to note that more than 60% of patients had negative feelings about self-sampling in the results of this study. The authors simply state that "This may reflect the Japanese personality (Page 18, line 302)" as one of the reasons. This sentence alone does not have enough grounds for the reason. It would be desirable to include suggestions on improving this result, which I believe will give readers more valuable tips.

3. The statement by the authors, "There would be women with negative impressions of the self-sampling HPV test in the general population, such women might also change their minds of experience.'' (Page 21, lines 359-361) I think this is an essential suggestion for the spread of self-sampling in Japan in the future. I think it would be more interesting for the readers if you mentioned it as a Finding instead of a Limitation and put forward proactive proposals for raising awareness of patients in Japan in the Discussion.

4. Overall, the text is large and redundant, so please try to keep it concise by omitting content that is not directly related to this research as much as possible. For example, it is not common for research papers targeting populations to develop an argument based on individual responses to the questionnaire (Page13, line219-223Pape21, line357-359). We also agree that the description about COVID-19 (page 21, line 367-372) is essential, but it is not directly related to the results of this study, so I recommend that you delete it.

Reviewer #2: This manuscript is a descriptive study(not RCT) to evaluate the acceptability and preference of self-sampling human papillomavirus(HPV) test.

The authors state that majority of the participants agreed with positive impressions about the self-sampling HPV test and they conclude this tool would help to raise screening uptake and detection of pre-cancer.

The HPV vaccine issue is a major public health challenge in Japan, and the author's awareness of the problem of evaluating self-sampling HPV tests to improve the effectiveness of screening is very important.

However, the response rate of the participants in this manuscript was very low. It therefore has an extremely large sampling bias. This bias could affect the conclusions of this manuscript and is fatal to the study.

Also, the purpose of this manuscript was to evaluate the acceptance and preference of self-sampling HPV test, which is difficult to evaluate because there was no comparison between the intervention group and the control group.

This study is a descriptive study, but the description of the study format is ambiguous.

Based on the above, we judge that this study is not of sufficient quality to be published in this journal.

The following are some of the issues to be addressed in detail.

1. The proportion of those who chose the self-sampling HPV test was very low (16.3%), which may cause sampling bias. Since the main purpose of this manuscript is to evaluate the positive impression of the self-sampling HPV test, this bias is a major problem that directly affects the conclusion of the manuscript, and is also insufficiently addressed in the limitation.

The table 1 describes only the background of the intervention group. It is possible that the intervention and non-intervention groups have different backgrounds.

2. Ⅼ225-227 states that older people tend to experience more difficulty regarding the pain of self-sampling HPV tests, while Ⅼ334-339 emphasizes that there is no association with age in the collection of HPV tests. The logic is inconsistent.

3.Table 2 and other tables have many items but are redundant and need to be made more compact. It would be desirable to consider changing to bar graphs, etc., which are visually easy to understand.

6. PLOS authors have the option to publish the peer review history of their article (what does this mean?). If published, this will include your full peer review and any attached files.

Reviewer #1: **Yes: **Eijiro Hayata

Reviewer #2: No

---

## [Author Response · Author response to Decision Letter 0]

2 Apr 2023

Responses to the Editor and Reviewers’ Comments

Submission ID: PONE-D-22-28147

Title: Acceptability of self-sampling human papillomavirus test for cervical cancer screening in Japan: A questionnaire survey in the ACCESS trial

We appreciate the Editor’s and Reviewers’ comments toward improving our paper. We have revised the manuscript to address all the comments, and our point-by-point responses are provided below. We have included the page and line numbers of the revisions made in the main text. Revisions in response to the Editor’s and Reviewers’ comments are highlighted in yellow in the revised manuscript with track changes.

Academic editor

Comment 1

English writing needs to be improved. Awkward or ambiguous English expressions and incorrect grammar are seen in current version, e.g. the 3rd sentence in Methods of Abstract, the last sentence in Results of Abstract, ... The authors are advised please to check thoroughly the entire manuscript for language improvement, and/or ask help for English editing if available at your own end.

Response 1

Thank you for your comment. Although we had asked an English editing company to edit our English, some incorrect expressions remained. We checked the entire manuscript and asked the company to re-edit our manuscript.

Comment 2

The current manuscript is unnecessarily lengthy. Please check the reviewers’ comments as well. Please write concisely in the revision with removing irrelevant sentences/paragraphs and duplicated/redundant materials.

Response 2

In line with comments from reviewers, we changed Table 2 to a graph (from page 12, lines 214–215) and deleted the descriptions based on individual responses in the Results section and those related to COVID-19 in the Discussion section. Additionally, some sentences in the Discussion section were deleted.

Comment 3

Although the parenting study is an ongoing control randomized trial, however the current manuscript analyzed a subset of the data from one arm only. Please make this crystal clear, specially in abstract and here there when it needs to be clarified.

Response 3

We added an explanation in the Abstract (page 3, lines 31–32) and revised the description in the Materials and Methods section (page 7, lines 126–129).

Comment 4

Please look at Reviewer 2’s comment about the sampling bias concern. Please define your study population explicitly and give the limitation where your findings can be generalized to.

Response 4

Thank you for your comment. In line with a comment from Reviewer 2, we revised Table 1 (from page 10, line 199, to page 11, line 204) and added explanations in the Discussion section (from page 19, line 352, to page 20, line 362). A major revision was to specify that the population of this study was women who had undergone the self-sampling HPV test, rather than the general population. Additionally, the Abstract (page 4, lines 49–50), the Statistical analysis section (page 9, lines 160–162), the Results section (from page 10, lines 195–197), and the Conclusion section (page 20, line 374) were also revised.

Reviewer 1

Comment 1

I have a question about patient selection criteria. Please explain briefly why you excluded patients who received regular cervical cancer screening and included only those who had not been screened for three years or more. We believe that asking patients who are regularly screened for their opinion on self-sampling is also helpful and has the advantage of reducing selection bias.

Response 1

Thank you for your comment. Self-sampling HPV testing has already been implemented in other countries, such as the Netherlands and Denmark, and the test has only been recommended for regular screening non-responders. Therefore, we assumed that a self-sampling HPV test, if implemented in Japan, would be performed by regular screening non-responders. We added this explanation from page 6, lines 110–112.

Comment 2

It is important to note that more than 60% of patients had negative feelings about self-sampling in the results of this study. The authors simply state that “This may reflect the Japanese personality (Page 18, line 302)” as one of the reasons. This sentence alone does not have enough grounds for the reason. It would be desirable to include suggestions on improving this result, which I believe will give readers more valuable tips.

Response 2

We agree with you and have deleted this sentence due to the lack of grounds for the reason. As already described in the text, to eliminate women’s concerns related to the accuracy of the self-sampling HPV test, it is necessary to provide information related to the self-sampling test in advance, such as the probability of invalid test results and the accuracy of the tests. We recently reported that the probability of invalid test results was rare among Japanese women; accordingly, we cited this article in the text (page 17, lines 303–305).

Comment 3

The statement by the authors, "There would be women with negative impressions of the self-sampling HPV test in the general population, such women might also change their minds of experience.'' (Page 21, lines 359-361) I think this is an essential suggestion for the spread of self-sampling in Japan in the future. I think it would be more interesting for the readers if you mentioned it as a Finding instead of a Limitation and put forward proactive proposals for raising awareness of patients in Japan in the Discussion.

Response 3

Thank you for your comment. We are glad that you recognize the importance of this study. In line with your comment, the descriptions were indicated in the Discussion section as a finding after the explanation of this study’s strengths. Additionally, a suggestion to promote the self-sampling test in Japan was added (from page 20, lines 364–372).

Comment 4

Overall, the text is large and redundant, so please try to keep it concise by omitting content that is not directly related to this research as much as possible. For example, it is not common for research papers targeting populations to develop an argument based on individual responses to the questionnaire (Page13, line219-223Pape21, line357-359). We also agree that the description about COVID-19 (page 21, line 367-372) is essential, but it is not directly related to the results of this study, so I recommend that you delete it.

Response 4

Thank you for your comment. We deleted the descriptions based on individual responses in the Results section and those related to COVID-19 in the Discussion section. Additionally, some sentences in the Discussion section were deleted.

Reviewer 2

Comment 1

The proportion of those who chose the self-sampling HPV test was very low (16.3%), which may cause sampling bias. Since the main purpose of this manuscript is to evaluate the positive impression of the self-sampling HPV test, this bias is a major problem that directly affects the conclusion of the manuscript, and is also insufficiently addressed in the limitation.

The table 1 describes only the background of the intervention group. It is possible that the intervention and non-intervention groups have different backgrounds.

Response 1

Thank you for your comment. We also recognize the selection bias in this study. In line with your comment, we revised Table 1 (from page 10, line 199, to page 11, line 204) and added explanations in the Discussion section (from page 19, line 352, to page 20, line 362). A major revision was to specify that the population of this study was women who had undergone the self-sampling HPV test, rather than the general population. Additionally, the Abstract (page 4, lines 49–50), the Statistical analysis section (page 9, lines 160–162), the Results section (from page 10, lines 195–197), and the Conclusion section (page 20, line 374) were also revised.

Comment 2

Ⅼ225-227 states that older people tend to experience more difficulty regarding the pain of self-sampling HPV tests, while Ⅼ334-339 emphasizes that there is no association with age in the collection of HPV tests. The logic is inconsistent.

Response 2

Thank you for your comment. We agree with you.

We would like to advocate that self-sampling HPV testing is an acceptable method regardless of age and duration without screening. Therefore, we deleted statements explaining that self-sampling HPV testing is more acceptable to the younger generation and revised the explanation in the Discussion section (page 19, lines 335–341). 

Comment 3

Table 2 and other tables have many items but are redundant and need to be made more compact. It would be desirable to consider changing to bar graphs, etc., which are visually easy to understand.

Response 3

According to your comment, we changed Table 2 to a graph (from page 12, lines214–215). The other tables remain unchanged because graphs would not allow for the presentation of some meaningful information.

---

## [Decision Letter · Decision Letter 1]

2 May 2023

PONE-D-22-28147R1Acceptability of self-sampling human papillomavirus test for cervical cancer screening in Japan: A questionnaire survey in the ACCESS trialPLOS ONE

Dear Dr. Fujita,

Thank you for submitting your manuscript to PLOS ONE. After careful consideration, we feel that it has merit but does not fully meet PLOS ONE’s publication criteria as it currently stands. Therefore, we invite you to submit a revised version of the manuscript that addresses the points raised during the review process.

We look forward to receiving your revised manuscript.

Kind regards,

Ruofei Du, PhD

Academic Editor

PLOS ONE

Journal Requirements:

Additional Editor Comments:

The manuscript needs additional clarifications before it can be accepted for publication.

Page 6, Line 107: It is unclear to define an inclusion criterion by stating “women considered the target population for cervical cancer screening by Ichihara City in 2021’. Please provide details on what this criterion really entails.

Page 8, Line 156: The numbers of participants falling into the category of ‘without registration’ are substantial as shown in different tables (Tables 1&2&3). The unknown composition of this category has the potential to bias the results. Could the authors provide information on the possible reasons that led to women not being registered and discuss how the study results would be affected under different scenarios?

Page 9, Statistical Analysis: Please add a description of how the trend test analysis was applied in the study.

Page 10, Line 195-197: There appears to be a lack of sufficient description and interpretation about the data presented in Table 1. For instance, although there is a statistically significant difference in the mean ages between the groups in Table 1, a difference of 44.1 vs. 44.7 may not be clinically meaningful. What are the implications of the differences in the distributions of age and duration without screening?

Page 12, Line 232-233: There are no 49.1% vs. 89.3% shown in Table 2. Should that be 46.9% vs. 86.7%?

Page 18, Line 315: Please rewrite this sentence ‘the knowledge was less in vulnerable women with lower screening rates’. Do you mean that women who had lower screening rates and are considered more vulnerable had less knowledge about HPV?

Reviewers' comments:

Reviewer's Responses to Questions

**Comments to the Author**

1. If the authors have adequately addressed your comments raised in a previous round of review and you feel that this manuscript is now acceptable for publication, you may indicate that here to bypass the “Comments to the Author” section, enter your conflict of interest statement in the “Confidential to Editor” section, and submit your "Accept" recommendation.

Reviewer #1: All comments have been addressed

Reviewer #2: (No Response)

2. Is the manuscript technically sound, and do the data support the conclusions?

Reviewer #1: Yes

Reviewer #2: Yes

3. Has the statistical analysis been performed appropriately and rigorously? 

Reviewer #1: Yes

Reviewer #2: Yes

4. Have the authors made all data underlying the findings in their manuscript fully available?

Reviewer #1: Yes

Reviewer #2: No

5. Is the manuscript presented in an intelligible fashion and written in standard English?

Reviewer #1: Yes

Reviewer #2: Yes

6. Review Comments to the Author

Reviewer #1: I would like to express my homage to the authors for sincerely responding to the reviewers' comments. I am convinced that this manuscript has been well revised and has an important message to clinicians. This paper is a significant contribution, and I recommend that it be accepted for publication.

Reviewer #2: The authors have addressed the issue of sampling bias as pointed out by the reviewers, and have also restructured the manuscript to make it more compact. In addition, the authors have revised the tables comparing participants who responded to the questionnaire with those who did not respond to. Based on the above, I concluded that this manuscript is acceptable.

7. PLOS authors have the option to publish the peer review history of their article (what does this mean?). If published, this will include your full peer review and any attached files.

Reviewer #1: No

Reviewer #2: No

---

## [Author Response · Author response to Decision Letter 1]

22 May 2023

Responses to the Editor’s Comments

Submission ID: PONE-D-22-28147

Title: Acceptability of self-sampling human papillomavirus test for cervical cancer screening in Japan: A questionnaire survey in the ACCESS trial

We appreciate the Editor’s comments toward improving our paper. We have revised the manuscript to address all the comments, and our point-by-point responses are provided below. We have included the page and line numbers of the revisions made in the main text. Revisions in response to the Editor’s comments are highlighted in yellow in the revised manuscript with tracked changes.

Academic editor

Comment 1

Page 6, Line 107: It is unclear to define an inclusion criterion by stating “women considered the target population for cervical cancer screening by Ichihara City in 2021’. Please provide details on what this criterion really entails.

Response 1

Thank you for your comment. The target population for cervical cancer screening provided by Ichihara City in 2021 was women whose age was an even number. The city set this eligibility requirement to offer cervical cancer screening once every two years to all women. We have added the relevant explanation in the manuscript (page 6, lines 111–114).

Comment 2

Page 8, Line 156: The numbers of participants falling into the category of ‘without registration’ are substantial as shown in different tables (Tables 1&2&3). The unknown composition of this category has the potential to bias the results. Could the authors provide information on the possible reasons that led to women not being registered and discuss how the study results would be affected under different scenarios?

Response 2

Thank you for your comment. As our descriptions were misleading, we have revised them and the changed the category’s name (page 8 lines 161–162). The category “no screening records” comprised women who had no records regarding cervical cancer screening provided by Ichihara City, such as the date of the screening. As described in the Discussion section (page 17, lines 294–300), Japan does not have a registry system for cervical cancer screening. Therefore, we could not truly determine screening non-responders and under-responders. Therefore, there is a possibility that all the categories pertaining to the duration without screening included women who had undergone the screening provided by other site, not the city. Explanations of the sensitivity analyses regarding this point are presented in the Statistical analysis (page 9, lines 174–177), Results (page 14, lines 263–265), and Discussion (from page 18, line 328–page 19, line 336) sections. 　

Comment 3

Page 9, Statistical Analysis: Please add a description of how the trend test analysis was applied in the study.

Response 3

We have added the explanation in the Statistical analysis section (page 9, lines 179–181).

Comment 4

Page 10, Line 195-197: There appears to be a lack of sufficient description and interpretation about the data presented in Table 1. For instance, although there is a statistically significant difference in the mean ages between the groups in Table 1, a difference of 44.1 vs. 44.7 may not be clinically meaningful. What are the implications of the differences in the distributions of age and duration without screening?

Response 4

We have described these differences in the Discussion section (page 20, lines 363–365). As you have mentioned, these differences may be trivial. However, because of the significant differences, we accepted the possibility that the sample included this study did not represent general population. This has been explained in the manuscript (page 20, lines 359–369).

Comment 5

Page 12, Line 232-233: There are no 49.1% vs. 89.3% shown in Table 2. Should that be 46.9% vs. 86.7%?

Response 5

Thank you for bringing this to our attention. Per your comment, we have corrected these numbers (page 12, line 238–page 13, line 239).

Comment 6

Page 18, Line 315: Please rewrite this sentence ‘the knowledge was less in vulnerable women with lower screening rates’. Do you mean that women who had lower screening rates and are considered more vulnerable had less knowledge about HPV?

Response 6

Thank you for your comment. We have revised the description as presented in page 18, lines 321–324. Additionally, we have cited more articles (reference numbers 38 and 39) in the revised manuscript.

---

## [Editor Report · Decision Letter 2]

26 May 2023

Acceptability of self-sampling human papillomavirus test for cervical cancer screening in Japan: A questionnaire survey in the ACCESS trial

PONE-D-22-28147R2

Dear Dr. Fujita,

We’re pleased to inform you that your manuscript has been judged scientifically suitable for publication and will be formally accepted for publication once it meets all outstanding technical requirements.

Kind regards,

Ruofei Du, PhD

Academic Editor

PLOS ONE
---

## [Editor Report · Acceptance letter]

30 May 2023

PONE-D-22-28147R2 

Acceptability of self-sampling human papillomavirus test for cervical cancer screening in Japan: A questionnaire survey in the ACCESS trial 

Dear Dr. Fujita:

I'm pleased to inform you that your manuscript has been deemed suitable for publication in PLOS ONE. Congratulations! Your manuscript is now with our production department. 

Kind regards, 

on behalf of

Dr. Ruofei Du 

Academic Editor

PLOS ONE